# Genome-Directed Cell Nucleus Assembly

**DOI:** 10.3390/biology11050708

**Published:** 2022-05-05

**Authors:** Sergey V. Razin, Sergey V. Ulianov

**Affiliations:** 1Institute of Gene Biology, Russian Academy of Sciences, 119334 Moscow, Russia; sergey.v.ulyanov@genebiology.ru; 2Faculty of Biology, M.V. Lomonosov Moscow State University, 119234 Moscow, Russia

**Keywords:** chromatin, genome folding, cell nucleus assembly, nuclear compartmentalization, liquid condensates, nuclear mechanics

## Abstract

**Simple Summary:**

Speckles and other nuclear bodies, the nucleolus and perinucleolar zone, transcription/replication factories and the lamina-associated compartment, serve as a structural basis for various genomic functions. In turn, genome activity and specific chromatin 3D organization directly impact the integrity of intranuclear assemblies, initiating/facilitating their formation and dictating their composition. Thus, the large-scale nucleus structure and genome activity mutually influence each other. The cell nucleus is frequently considered a compartment in which the genome is placed to protect it from external forces. Here, we discuss the evidence demonstrating that the cell nucleus should be considered, rather, as structure built around the folded genome. Decondensing chromosomes provide a scaffold for the assembly of the nuclear envelope after mitosis, whereas genome activity directs the assembly of various nuclear compartments, including nucleolus, speckles and transcription factories.

**Abstract:**

The cell nucleus is frequently considered a cage in which the genome is placed to protect it from various external factors. Inside the nucleus, many functional compartments have been identified that are directly or indirectly involved in implementing genomic DNA’s genetic functions. For many years, it was assumed that these compartments are assembled on a proteinaceous scaffold (nuclear matrix), which provides a structural milieu for nuclear compartmentalization and genome folding while simultaneously offering some rigidity to the cell nucleus. The results of research in recent years have made it possible to consider the cell nucleus from a different angle. From the “box” in which the genome is placed, the nucleus has become a kind of mobile exoskeleton, which is formed around the packaged genome, under the influence of transcription and other processes directly related to the genome activity. In this review, we summarize the main arguments in favor of this point of view by analyzing the mechanisms that mediate cell nucleus assembly and support its resistance to mechanical stresses.

## 1. The Folded Genome Instructs the Assembly and Determines the Cell Nucleus’s Shape and Rigidity

It is well known that the assembly of the cell nucleus after mitosis is nucleated by decondensing chromosomes [1]. Lamin B binds to lamina-associated chromatin domains (LADs) before a complete lamina is formed [2]. The components of the pore complex also bind to chromatin, creating nucleation centers for the subsequent assembly of functional pore complexes [3]. Many membrane proteins, including LBR, nucleoporins, and NDC1, bind directly to DNA during decondensation of metaphase chromosomes [1]. The size and shape of the nucleus are largely determined by the genome size and the genome packaging’s mode. The nuclear lamina is frequently considered a mechanical scaffold for the nucleus [4]. However, by itself, it is relatively elastic; therefore, it is unable to maintain the shape of the nucleus after digestion of chromatin by nucleases [5]. Chromatin, however, does have sufficient mechanical rigidity [5]. Changing the way chromatin is packaged by modulating histone modification profiles results in a change in nuclear rigidity [6,7]. Such changes occur, in particular, under the influence of mechanical stress, when it becomes necessary to reduce the rigidity of the nucleus in order to prevent its destruction [8]. This reduction is achieved by decreasing the amount of nuclear lamina-associated heterochromatin [8].

Mutations in the genes’ encoding lamins are the cause of pathologies termed “laminopathies” [9,10]. These pathologies are associated with a violation of the nucleus’s mechanical strength. The nuclei become more deformable and less resistant to mechanical strain, frequently resulting in nuclear rupture, blebbing, and other injures. At first sight, these changes directly point to the nuclear lamina as a structure that ensures the nucleus’s strength. However, the effect of mutations causing laminopathies is rather complex and also involves changes in chromatin folding [11]. Thus, in Hutchinson–Gilford progeria syndrome, constitutive and facultative heterochromatin is partially lost [7,12,13,14]. An increase in the level of heterochromatinization by treating cells with a histone demethylase inhibitor leads to the restoration of Hutchinson–Gilford progeria syndrome patients’ cell nuclei morphology [7], further indicating the important role of heterochromatin in ensuring the strength of the cell nucleus.

The overall construction of the cell nucleus is largely determined by the fixation of chromosome territories on the nucleolus and nuclear lamina and the presence of the interchromatin compartment (IC), in which various functional compartments (nuclear bodies) are assembled [15,16]. The emergence of these compartments is directly determined by the activity of various parts of the genome and, in many cases, is a consequence of the physicochemical process of liquid phase separation [17,18]. The mechanisms that ensure the existence of the interchromatin compartment remain less clear. There are reasons to believe that an important role is played here by the fact that active chromatin is located on the surface of interchromatin channels [19], whereas the channels themselves are filled with RNA in complex with proteins [20]. It has been proposed that this RNA plays a key role in maintaining the architecture of chromosome territories [21]. The possibility that the content of interchromatin channels represents a distinct phase-separated liquid domain within the cell nucleus also deserves consideration [20].

## 2. Active and Inactive Neighborhoods in the Cell Nucleus

It is well known that there are inactive regions in the nucleus in which a significant part of heterochromatin is concentrated. These are the perinucleolar and lamina-associated compartments [22,23,24,25]. Recent studies show that there are also active hubs located near speckles [25,26,27,28]. The formation of these functional nuclear domains is directed by the genome, particularly through the transcription of repeat sequences. It is well known that in mammalian cells, euchromatin is enriched in SINE type repeats, whereas heterochromatin is enriched in LINE-type repeats. Recently, LINE transcription in embryonic stem cells has been shown to be essential for the formation of lamina-associated domains (LADs) [29,30]. Similarly, transcripts of minor and major satellite DNA regions in mouse ES cells are essential for the organization of chromocenters, whereas the transcription of Kcnq1ot1 triggers the assembly of inactive chromatin domain, which includes several imprinted genes [26]. The formation of an inactive zone around the nucleolus is associated with the presence of inactive copies of ribosomal genes in this area [31]. In each nucleolus, only a portion of rDNA repeats is active. Silencing of other rDNA repeats is mediated by the nucleolar remodeling complex (NoRC), which recruits histone-modifying and DNA-methylating enzymes [32,33]. Apparently, the NoRC complex recruited to the perinucleolar layer may also target other chromatin regions that happen to be in the vicinity. Centromeric heterochromatin is frequently located close to the nucleolus, and it has been shown that the depletion of TIP5, a subunit of the NoRC complex, compromises both rDNA silencing and centromeric heterochromatin assembly [34].

## 3. Transcription-Coupled Assembly of Functional Nuclear Compartments

The cell nucleus contains a large number of distinct functional compartments that are formed with the participation of newly synthesized transcripts and are localized near the sites of the synthesis of these transcripts. These compartments include the nucleolus, Cajal bodies, paraspeckles, histone gene loci and activating compartments assembled on enhancers. The processes of assembly of all these compartments have much in common. The newly synthesized RNA acts as a scaffold, to which proteins are attracted to form a liquid-phase condensate [20]. This condensate can accumulate passenger proteins that perform various functions, including (i) transcription activators, (ii) transcription repressors and (iii) RNA processing factors. Due to its size and sometimes because of anchorage on DNA through the transcribed RNA, the condensate remains close to the scaffolding RNA transcription site. Below, we briefly discuss the assembly of the above-mentioned compartments.

### 3.1. The Nucleolus as a Multicomponent Phase Condensate

The nucleolus is the largest and most recognizable functional compartment within the cell nucleus. The assembly of the nucleolus is directed by the transcription of ribosomal genes [35]. In mammals, the nucleolus is formed around tandem ribosomal DNA repeats located at nucleolar organizer regions (NORs) on acrocentric chromosomes. The integrity of the nucleolus is maintained only under conditions of active transcription of rRNA [35]. Under physiological conditions, the nucleolus has a tripartite structure and consists of three well-defined compartments: the fibrillar center (FC), the dense fibrillar component (DFC), and the granular component (GC). As a rule, several fibrillar centers are present in the nucleolus, each of which is surrounded by a DFC. The newly synthesized ribosomal RNA acts as a scaffold for the formation of DFC and GC, attracting first nucleolin and then (after removal of introns), nucleophosmin. Both of these proteins are capable of forming phase condensates that do not mix with each other [35]. Inactive ribosomal genes located on the periphery of the nucleolus act as nucleation centers to create a repressive chromatin domain [36]. This domain constitutes an anchoring point for chromosomal territories via nucleolar-associated regions (NADs) [37,38], which significantly overlap with LADs [39,40]. Being attached to the nucleolus and nuclear lamina, the chromosomal territories become stretched between the nuclear envelope and nuclear interior.

### 3.2. Liquid Condensates Containing RNA Polymerase II

Pioneering work in the laboratory of Peter Cook demonstrated that in the cell nucleus, RNA polymerase II is organized into clusters, which were termed “transcription factories” [41,42,43]. Currently, there is strong evidence that these clusters are predominantly formed by LLPS and are liquid-phase condensates containing RNA polymerase II itself, a mediator, transcription factors and other proteins involved in transcription [44]. Additionally, nuclear actin polymerization induced by phase separation of RNAPII, with positive regulator of actin polymerization N-WASP, is essential for the serum- and IFN-γ-induced RNAPII clustering [45]. Together with the evidence that inhibition of myosin VI activity disrupts RNAPII clusters [46], these data point to a role of actin/myosin machinery in the formation/maintenance of transcription-related nuclear assemblies. However, the idea that transcription is carried out by immobilized RNA polymerase II has not received sufficient experimental support. The results of recent studies indicate that liquid-phase condensates containing RNA polymerase II are assembled on enhancers [47,48,49] and contain predominantly RNA polymerase II molecules that are not involved in elongation. eRNA transcribed from active enhancers might serve as a scaffold for the assembly of these condensates [50,51]. Promoters are recruited to such condensates, and pre-initiation complexes are assembled inside the condensates. After the start of elongation, the transcription complexes leave the condensate [52,53,54].

### 3.3. Liquid-Phase Condensates Containing Enzymes Involved in Primary Transcript Processing

The cell nucleus contains many different functional compartments involved in the processing of newly synthesized transcripts. These compartments include, in particular, speckles, paraspeckles, Cajal bodies and histone locus bodies [17,18]. All of these compartments are assembled in close proximity to transcription sites of RNAs that are processed within the compartments. In those cases where functional compartments are involved in the processing of RNA from several genomic loci (for example, in the processing of histone RNA transcribed from two histone gene loci on mouse chromosome 13), these loci become positioned in spatial proximity within the cell nuclei [26]. A specific steric problem arises in the case of splicing speckles because genes containing introns are distributed throughout the genome. Apparently, speckles are formed next to the most actively transcribed genes, and a significant number of genes located in the neighborhood are attracted to each of the speckles [26,55]. According to one of the hypotheses, the need to attract active genes to speckles contributed to the evolution of the cluster organization of genes manifested in the appearance of active and inactive segments in chromosomes (R and G bands) [56].

## 4. Modulation of Activity and Spatial Organization of the Genome through Mechanical Stress

As a platform for assembling the cell nucleus, the folded genome is relatively sensitive to mechanical impacts on the nucleus. The consequences of mechanical impacts on the cell nucleus can be divided into two categories: (i) changes in the manner of chromatin folding, aimed at adapting the architecture of the nucleus to external influences, and (ii) changes in transcription profiles resulting from the 3D genome’s reconfiguration. Recent evidence suggests that the folded 3D genome as a whole represents a viscoelastic rheological element of the nucleus that can absorb mechanical shocks. The rheological properties of a folded genome are mainly determined by heterochromatin and may be modified by heterochromatin assembly/disassembly [8,57,58,59]. It should be mentioned that 3D genome folding is rather flexible [60,61,62,63]. While adsorbing mechanical stresses, chromatin moves and changes its local configurations [64]. Considering that regulatory contacts between enhancers and promoters are established at the 3D genome level and that genome folding is connected to nuclear organization [65,66], it is logical to expect that mechanical stresses on the nucleus will cause changes in transcription profiles. Indeed, the ability to activate/suppress the work of various genes through mechanical effects on the nucleus has been demonstrated in a number of studies [67,68,69,70,71,72]. Although, in many cases, changes in transcription profiles and chromatin folding occur because of the activation of mechanosensitive nuclear pores and ion channels [8,59,73,74,75], the direct effect of external mechanical forces on transcription via modulation of chromatin folding and chromosome positioning within the cell nucleus has also been reported [69,75,76,77].

## 5. Concluding Remarks

The cell nucleus is a complex structure, divided into functional zones and harboring many functional compartments, the most significant of which is the nucleolus. The question of whether there is some kind of structure that supports the system of intranuclear compartments has been discussed for many years. The results of the work on chromatin solubilization have been interpreted in terms of the existence of a proteinaceous nuclear skeleton (nuclear matrix). One of the arguments in favor of such an interpretation is that after DNA digestion and histone extraction, nuclei retain their shape and some morphological features [78]. Within the framework of the nuclear matrix concept, the nucleus is considered as a kind of box with sections in which the genome is located in a specific manner [79,80]. Subsequent studies have not provided convincing evidence for the existence of a proteinaceous nuclear matrix in living cells [81]. At the same time, it has become clear that the organization of the cell nucleus is highly mobile and directly reflects the activity of the genome, whereas the cell nucleus’s mechanical properties are largely determined by the physical characteristics of the genome packaged in chromatin. In this review, we presented arguments in favor of a model assuming that the folded genome is the structural basis of the cell nucleus (Figure 1). This model concerns both the mechanical properties (shape and resistance to mechanical stress) and functional compartmentalization of the nucleus. The implementation of genome activity, primarily transcription, directs the formation of functional domains and compartments of the nucleus, including the so-called nuclear bodies. At the same time, the formation of nuclear functional compartments contributes to the establishment of spatial contacts between different parts of the genome (i.e., enhancer-promoter contacts, polycomb bodies, association of active genes with nuclear speckles). Thus, there is a close relationship between the formation of the cell nucleus and the genome’s spatial organization. Following this logic, it would be reasonable to state that the folded genome is organized into the cell nucleus, rather than just localized within the nucleus.

## Figures and Tables

**Figure 1 biology-11-00708-f001:**
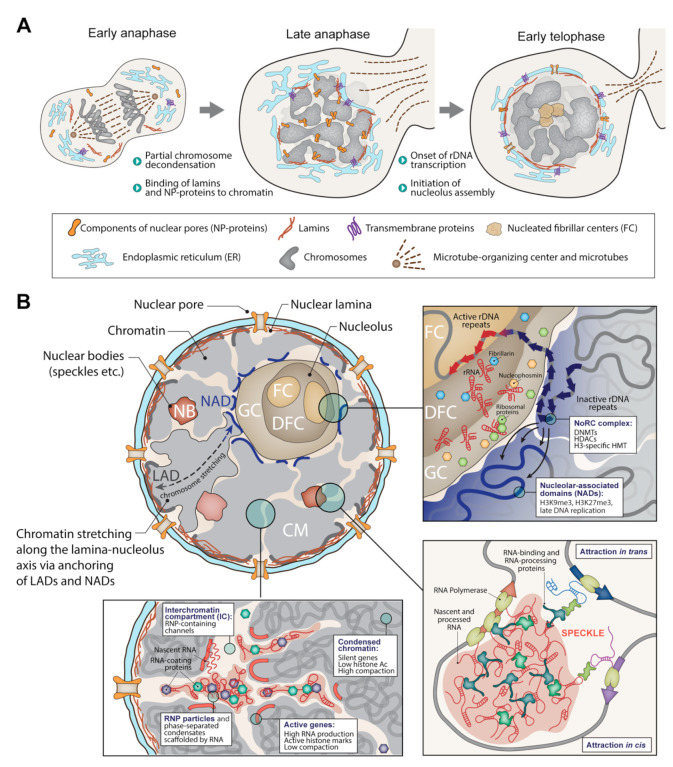
Genome-centric view of the nucleus assembly. (**A**) Chromosome decondensation at late anaphase initiates the assembly of the nuclear lamina and pore complexes through the binding of lamins and nuclear pore (NP) proteins to chromatin. The onset of the rDNA transcription at early telophase results in the formation of fibrillar centers. Along with the assembly of the nuclear envelope, this establishes the nuclear lamina-nucleolus axis along which chromosomes are stretched via anchoring of lamina-associated domains (LADs) and nucleolus-associated domains (NADs). (**B**) The folded genome serves as a structural basis for the nucleus structure. Active transcription of the rDNA repeats located in between the fibrillar centers (FC) and dense fibrillar component (DFC) is the prerequisite for the nucleolus integrity. Inactive rDNA repeats are located at the surface of the granular component (GC) and are silenced by the NoRC complex recruiting DNA-methyltransferases (DNMTs), histone deacetylases (HDACs), and H3K9- and H3K27-specific histone methyltransferases. Increased local concentration of these enzyme complexes promotes a repressive chromatin state of the genome loci located at the nucleolus within the NADs. At the nucleus periphery, chromosomes are anchored to the lamina within LADs. This results in the stretching of the bulk chromatin mass (CM) along the NAD–LAD axis. Within the chromosome territories, actively transcribed genes nucleate the formation of various transcription-related bodies such as speckles. These structures are predominantly formed by the liquid–liquid phase separation, driven by numerous weak interactions between RNA and RNA-binding proteins. Recruitment of distant genes to these bodies is one of the determinants of chromosome territory folding. Produced RNAs in complex with RNA-binding proteins accumulate and migrate towards the nuclear pores, forming interchromatin compartment (IC), the network of “channels” penetrating chromosome territories.

## Data Availability

Not applicable.

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
