# Peer review of "Genome-Directed Cell Nucleus Assembly"

_biology, 2022, doi:10.3390/biology11050708_

Round 1

Reviewer 1 Report

This review article by Sergey V. Razing and Sergey V. Ulianov describes genomic mechanisms implicated in the cell nucleus assembly. The review is narrow-focused and well illustrated, and will bring the attention of the specialists working in the field. I have only two minor remarks:

  1. "...for example, in the processing of histone RNA transcribed from two histone gene loci on chromosome 13" - authors should indicate which species they are referring to. In general, if there are taxon-specific mechanisms described in the manuscript, I suggest mentioning the taxon explicitly.
  2. The authors dedicated separate subsections to describe active nuclear compartments, i.e. the nucleolus, "Liquid condensates containing RNA polymerase II" and "Liquid-phase condensates containing enzymes involved in primary transcript processing". It is not clear why authors did not cover heterochromatin condensates as a separate section?

Overall, the review quality is high, and I recommend accepting it after minor revision.

Author Response

  1. "...for example, in the processing of histone RNA transcribed from two histone gene loci on chromosome 13" - authors should indicate which species they are referring to. In general, if there are taxon-specific mechanisms described in the manuscript, I suggest mentioning the taxon explicitly.

Reply: In the revised version of the MS we have indicated that in this sentence we spoke about mouse chromosome 13 (line 158 of the revised MS).

  1. The authors dedicated separate subsections to describe active nuclear compartments, i.e. the nucleolus, "Liquid condensates containing RNA polymerase II" and "Liquid-phase condensates containing enzymes involved in primary transcript processing". It is not clear why authors did not cover heterochromatin condensates as a separate section?

Reply: The aim of this opinion article is to show how folded 3D genome directs assembly of the cell nucleus. We do not address the actual mechanisms of genome folding, as this topic has been covered in a number of recent review articles, including our own (Kantidze and Razin, Nucleic Acids Res, 2020) and these mentioned by the third reviewer. In accord with this logic, in the section 3 of the MS we discuss functional nuclear compartments that are located outside chromatin domain (i.e., in interchromatin channels). Discussion of the assembly of either euchromatin or heterochromatin is beyond the scope of this opinion article.  

Reviewer 2 Report

The opinion piece by Razin and Ulianov is very interesting and thought provoking. The basis is drawn from clear evidence/rationale. I would be interested to see a mention to the roles/impact of nuclear actin and myosin within their proposal  - especially with the recent evidence for roles of nuclear myosin and actin in transcription organisation: https://www.nature.com/articles/s41467-022-28962-w and here https://www.science.org/doi/10.1126/sciadv.aay6515 

Author Response

Indeed, action of the nuclear actin/myosin machinery in relation to the nucleus compartmentalization is obviously underestimated to date. Following the reviewer’s suggestion, we cited these recent articles in the revised version of the MS (lines 135-142): “Also, nuclear actin polymerization induced by phase separation of RNAPII with positive regulator of actin polymerization N-WASP, is essential for the serum- and IFN- g-induced RNAPII clustering [45]. Together with the evidence that inhibition of myosin VI activity disrupts RNAPII clusters [46], these data point at a role of actin/myosin machinery in the formation/maintenance of transcription-related nuclear assemblies.”

Reviewer 3 Report

In this submitted manuscript, authors presented a review of mechanisms that mediate cell nucleus assembly and stated that there is a close relationship between the formation of the cell nucleus and the genome’s spatial organization. Although it is not new, in previous studies it has been showed that genomes are not randomly arranged within the nucleus (Genome Organization and Function in the Cell Nucleus, K. Rippe, Ed. (Wiley-VCH Verlag, Weinheim, Germany, ed. 1, 2011)). In spite of not being particularly original, the manuscript presents previous results and methods in a rational and convincing manner, but the resolution of the figure could be improved.

In my opinion, this review does not offer any new results in comparison to other more thorough articles on the topic (E. H. Finn, T. Misteli, Science 365, eaaw9498 (2019). DOI: 10.1126/science.aaw9498). Even though the authors described it as a review, the manuscript was classified as an opinion. I selected the item "Reject" based on the manuscript itself but leave it up to the editor to decide whether a mini review should be included as an "opinion".

Author Response

We do agree that our MS is not a full-scale review of research on the organization of the three-dimensional genome and the functional impact of this organization. However, we do not agree that our MS opinion does not offer new results compared to previously published papers. In an excellent review published by Finn and Misteli, which was mentioned by a reviewer, genome-directed assembly of the cell nucleus is not considered at all. The authors discuss the mechanisms of genome folding, paying special attention to the influence of stochastic processes. Obviously, the topic of our opinion article is fundamentally different. Furthermore, this topic was not yet properly addressed in published reviews.

Round 2

Reviewer 3 Report

The revised version of the manuscript does not present any major changes. As the authors stated this is not a full-scale review, I will stick to my previous opinion and let the editor make the final decision. 

Author Response

As explained in the answer to the first review of this reviewer, we did not intend to present a detailed analysis of the current state of the 3D genome studies. Our goal was rather to outline new ideas about the role of the 3D genome as a platform for the assembly of the cell nucleus. That is why we designated our MS as an Opinion article.